# Personality Risk Factors for Vape Use amongst Young Adults and Its Consequences for Sleep and Mental Health

**DOI:** 10.3390/healthcare12040423

**Published:** 2024-02-06

**Authors:** Simon L. Evans, Erkan Alkan

**Affiliations:** 1Faculty of Health and Medical Sciences, School of Psychology, University of Surrey, Guildford GU2 7XH, UK; 2Faculty of Arts and Social Sciences, School of Psychology and Counselling, Open University, Cardiff CF10 1AP, UK; erkan.alkan@open.ac.uk

**Keywords:** e-cigarette, vape, nicotine, personality, traits, mindfulness, chronotype, rumination, self-compassion, sleep, mental health, anxiety, depression, alcohol, loneliness, young adults, students

## Abstract

(1) Background: The surge in vape (e-cigarette) use among young adults is concerning, as there is limited knowledge about risk factors and health consequences. This study explores the personality traits linked to vape use, and associations between vaping and chronotype, sleep quality, and mental health, among young adults. (2) Methods: 316 participants, aged 18–25, completed measurements of mindfulness, rumination, self-compassion, anxiety/depression, chronotype, and sleep quality. (3) Results: the vape user group scored significantly lower on mindfulness, higher on rumination, and lower on self-compassion. Vape users were more likely to be evening types and had significantly lower sleep quality and higher anxiety symptoms, as well as higher alcohol use and loneliness (at trend) (4) Conclusions: These novel findings enhance our understanding of what might predispose young adults to vaping and the potential impact on their mental health and sleep quality. Findings point to specific cognitive/personality traits as vaping risk factors, which could inform intervention strategies.

## 1. Introduction

Electronic cigarettes (e-cigarettes or ‘vapes’) are battery-operated products that use a nicotine-containing solution to deliver nicotine via inhalable aerosol, although nicotine-free solutions are also available. Vape use, amongst youth in particular, has risen drastically over the last decade. Data from the United States, Canada, and England in 2019 suggests that over 30% of young people aged 16–19 are regular vape users [1]. In the UK, vape use is highest among young adults, with prevalence rising year on year (15.5% in 2022 compared with 11.1% in 2021); this is particularly evident among females, with young adult female usage more than tripling between 2021 and 2022 [2]. This has raised concern, as little is known regarding the health consequences of long-term use. Carcinogens and other toxins in e-cigarettes may well impact physical health [3], but also of concern is the possible impact on mental health. A recent systematic review found that youth vaping has been associated with various serious mental health diagnoses [4]. Critical aspects of brain maturation are regulated by nicotinic acetylcholine receptors (nAChRs). Disruption of cholinergic systems during this time with nicotine, via tobacco or e-cigarettes, has adverse consequences for adolescent brain development, inducing structural and neurochemical alterations that impact neural function and may induce epigenetic changes that increase the risk of further substance abuse [5]. However, youth often misunderstand or misperceive vaping. Evidence suggests that many do not know that nicotine is a tobacco derivative; the flavouring commonly found in vapes enhances their appeal, with US and UK adolescents erroneously believing that fruit-flavoured vapes are less harmful [6,7]. Thus, given their rising popularity amongst young adults in particular, it is important that we gain a more thorough understanding of the inter-individual factors that can increase the likelihood of a young person taking up vape use, as well as a fuller understanding of its consequences for physical and mental health.

Various studies have examined links between vape use and mental health, and data shows there is a greater chance of anxiety, stress, depression, and substance use in vape-using young adults [8,9], as well as higher levels of perceived stress [10]. However, as indicated by a recent review, not all studies have found positive associations; three of six cohort studies found associations with depression, while only four looked at links to anxiety, and findings are not clear-cut [11]. However, the review notes that some studies did not use comprehensive mental health measurement scales, which hampers interpretation. Even fewer studies have explored the psychological factors that predict vape use and susceptibility in young adults. Based on findings from studies on combustible cigarette use, it could be hypothesised that four of the Big Five factors (i.e., extraversion, neuroticism, conscientiousness, and agreeableness) might predict e-cigarette use. One study suggests that those lower in conscientiousness and higher in neuroticism are more likely to be vape users [12]. However, in another study, none of those variables were predictors, suggesting that perhaps the Big Five does not generalise from combustible cigarette use to vape use [13]. Thus, in the current study, we focus instead on some other potentially relevant traits as factors: trait mindfulness, rumination, and self-compassion. Studies suggest that young people may smoke to self-regulate negative feelings such as stress, depression, and anger [14]. Indeed, young adults with low self-control and poorer emotion regulation [15] are more likely to vape. Mindfulness, an enhanced attention to and awareness of present-moment experience [16], could represent a protective characteristic as it appears to be positively associated with affect regulation competencies [16]. Indeed, studies suggest that trait mindfulness has a negative relationship with cigarette smoking frequency [17] and dependence [18]. However, this relationship has not been tested in relation to vaping. Rumination (perseverative cognition focused on negative content) is associated with various unhealthy behaviours, including smoking [19]. This is unsurprising given that rumination results in emotional distress, and the reduction and regulation of negative affect is a major motivator of nicotine use [14]. Again, however, potential links between ruminative tendencies and vape use have not been investigated to date. Here, we also consider the role of self-compassion (mindful awareness of oneself, and treating oneself kindly during difficult and challenging experiences), since self-compassion is associated with health-promoting behaviours such as reduced cigarette smoking [20], as well as better emotion regulation [21], but again has not been studied in relation to vape use.

Another important factor that needs exploring in relation to vape use is that of chronotype. Chronotype, or circadian/diurnal preference, refers to individual differences in activity-rest cycles over the 24-hour period [22]. Eveningness, a preference for a later sleep period and activity in the evening, is known to be connected with higher stimulant consumption (e.g., alcohol, caffeine, and tobacco) as well as poorer mental health [23]. A recent review found robust links with smoking but noted a lack of studies on vape use [24]. A later sleep time might provide an additional opportunity to vape, contributing to more frequent use and dependence. Therefore, we here test the hypothesis that vape use is more prevalent amongst evening types. The effect of vaping on sleep quality also needs more exploration. The stimulant properties of nicotine may give rise to sleep difficulties, such as falling or staying asleep. Indeed, cigarette users report less total sleep and higher insomnia symptoms [25]. In a sample of adult vape users, greater vape dependence was related to poorer subjective sleep quality, although 75% of the sample were also cigarette smokers [26]. Another large study amongst college students found that both cigarette and vape users reported significantly more sleep difficulties than never users, and vape users reported greater sleep medication use than cigarette users [27]. The relationship may be bidirectional, however, as a study in adolescents showed that sleep deprivation increases susceptibility to the initiation of vape use, perhaps as a compensatory strategy [28].

To address the inconsistencies and knowledge gaps outlined above, the current study sets out to better characterise the personality risk factors for vape use amongst young adults. We also explore how vape use associates with mental health symptoms, sleep quality and chronotype, loneliness, and alcohol use. We predicted that the vape-user group would present with lower trait mindfulness and higher rumination. Although this has not been investigated previously, there is a strong theoretical rationale for this prediction. Likewise, we predicted that vape use will be associated with evening chronotype and lower sleep quality, addressing a paucity of previous work in relation to these. Also, we examined whether vape use is associated with poorer mental health (depression and anxiety symptoms), as the existing literature is inconsistent. We also included a measure of loneliness, as no previous work has investigated links to this, although there is some evidence that combustible cigarette smoking and loneliness are related [29]. Higher alcohol use was also predicted, as previous studies show higher levels of alcohol consumption among vape users. 

## 2. Materials and Methods

### 2.1. Participants

Participants were undergraduates aged 18–25 at the University of Surrey who completed the survey during a lecture (opportunity sampling). The study was approved by the University of Surrey Ethics Committee. N = 316 individuals completed the survey (61 males, 245 females, and 10 preferred not to say). The mean age was 20.0 (SD = 1.5 years).

### 2.2. Measures

#### 2.2.1. Five-Facet Mindfulness Questionnaire (FFMQ)

The Five Facets of Mindfulness Questionnaire Short Form (FFMQ-SF) was used to measure mindfulness levels [30]. The questionnaire contains 24 items with 4 or 5 questions for each facet of mindfulness: observing, describing, acting with awareness, non-judging inner experience, and non-reactivity to inner experience. The FFMQ was chosen because it is a unifying and comprehensive scale that integrates the conceptualizations of mindfulness from five previous (validated) mindfulness scales and has high levels of validity and reliability [31]. 

#### 2.2.2. Hospital Anxiety Depression Scale (HADS)

The Hospital Anxiety and Depression Scale (HADS) [32] was used to measure symptoms of depression and anxiety and consists of 14 items. Half of the items measure depressive symptoms, and the other half measure anxiety. Participants choose one response from four based on how they’ve been feeling the past week. Scoring for each item ranges from zero to three, with three denoting the highest anxiety or depression level. A total subscale score of 8 or above (out of a possible 21) indicates clinically significant symptoms of anxiety or depression. These cut-offs show good clinical validity, including in student samples [33].

#### 2.2.3. Pittsburgh Sleep Quality Index (PSQI)

Subjective sleep quality was measured by the Pittsburgh Sleep Quality Index (PSQI) [34]. The PSQI is comprised of 7 components for assessing sleep quality: subjective sleep quality, sleep latency, sleep duration, sleep efficiency, sleep disturbances, use of medication, and daytime dysfunction. The total PSQI score ranges from 0 to 21, with higher scores indicating poorer sleep quality. The diagnostic accuracy of the PSQI to detect insomnia is very high, with a threshold of >5 recommended for young adults [35].

#### 2.2.4. Rumination Reflection Questionnaire-Rumination Subscale (RRQ-R)

Rumination was measured using the 12-item rumination subscale of the Rumination-Reflection Questionnaire (RRQ) [36]. Participants indicate their agreement with statements such as ‘I spend a great deal of time thinking back over my embarrassing or disappointing moments’ on a 5-point Likert scale; higher scores indicate greater ruminative tendencies. High levels of validity and reliability have been demonstrated [36].

#### 2.2.5. Reduced Morningness-Eveningness Questionnaire (rMEQ)

Diurnal preference was measured using the 5-item Reduced Morningness-Eveningness Questionnaire (rMEQ) [37]. This measure instructs participants to rate their responses on a 4-point scale (e.g., “During the first half-hour after you wake up in the morning, how do you feel?” “Very tired”, “fairly tired”, “fairly refreshed”, and “very refreshed”). Lower scores indicate a greater evening preference. Scores range from 4 to 25: scoring 4–11 indicates eveningness, 12–17 indicates intermediate-type, and 18–25 indicates morningness.

#### 2.2.6. De Jong Gierveld Loneliness Scale 

Loneliness was assessed using the 6-item De Jong Gierveld Loneliness Scale, a well-used and validated instrument for loneliness. Satisfactory reliability (α = 0.70–0.76) has been shown for this scale [38]. Scores were summed, and a higher total indicated higher loneliness.

#### 2.2.7. Self-Compassion Scale (SCS) 

Self-compassion was assessed using the Self-Compassion Scale (SCS) [39], which measures the three main components of self-compassion: self-kindness versus self-judgement, common humanity versus isolation, and mindfulness versus over-identification. Moreover, 12 items are rated on a scale of 1 to 5. Higher total scores indicate higher self-compassion.

#### 2.2.8. Vaping/Cigarette Use

Participants were asked to answer (with yes or no) the question, “Do you use e-cigarettes/vape?” They also indicated their cigarette use by answering, “Approximately how many cigarettes do you smoke per week (including weekends)?” and alcohol use by answering, “Approximately how many units of alcohol (a unit is approximately equivalent to a small glass of wine or half a pint of beer/lager) do you drink per week (including weekends)?”

Those who indicated smoking more than 5 cigarettes a week were classified as cigarette smokers.

### 2.3. Procedure

The survey was administered using the Qualtrics online platform. After consenting to take part, all participants completed the measures in the same order.

### 2.4. Statistical Analysis

All data were analysed using the Statistical Package for Social Science (SPSS) 27.0. Groups were defined according to smoking or vaping status. Descriptive statistics were calculated, followed by assumption checks for parametric testing (Skewness/Kurtosis, Levene’s homogeneity-of-variance test). These indicated that assumptions of normality and homogeneity of variance were valid for these data, so a one-way ANOVA was used to test for differences between groups on the measures collected (IV: Vape use, DVs: FFMQ, Rumination, Diurnal preference, Self-compassion, HADS–Anxiety, HADS–Depression, Sleep Quality, Loneliness, and Alcohol use (units/wk).

## 3. Results

Amongst the sample (N = 316), 263 reported not using either cigarettes or vaping; 4 reported using cigarettes only; and 49 reported being current vape users (of which 14 were also cigarette users). Due to the low number of cigarette-only users, these were excluded to create two groups: a control group of N = 263 (comprising 201 females (76%), 54 males (21%), and 8 preferred not to say (3%)); and a vape-user group of N = 49 (comprising 41 females (84%), 6 males (12%), and 2 preferred not to say (4%)). There was no significant difference in the gender distribution between groups, based on a chi-square test. Between-group differences on the measures were tested using ANOVA; results and descriptive statistics are reported in Table 1. The two groups were closely matched in terms of age.

### 3.1. Between-Group Tests on Trait Characteristics

The vape-user group was found to have significantly lower levels of trait mindfulness as measured by the FFMQ, as well as significantly higher levels of rumination. The vape-user group also scored significantly lower on self-compassion. In terms of diurnal preference, the vape-user group scored significantly higher on the rMEQ, indicating a greater tendency towards evening preference. To explore this further, we applied the rMEQ categorical cut-off of <12 for evening-type to the data. This showed that in the control group, 39.9% were categorised as evening types, while in the vape-user group, 73.5% were categorised as evening types. This illustrates the extent to which evening types were overrepresented in the vape-user group.

### 3.2. Between-Group Tests on Mental Health Measures

The vape-user group was found to have significantly higher levels of anxiety as measured by the HADS, but no effect was observed for depression. To explore this further, we applied the HADS categorical cut-off of 8 or above for clinically significant levels of symptoms to the data. This showed that in the control group, 77.6% were categorised as having significant anxiety symptoms, while in the vape-user group, 95.9% were categorised as having significant anxiety symptoms. In contrast, on depression, in the control group, 47.5% were categorised as having significant depression symptoms, while in the vape-user group, 51.0% were categorised as having significant depression symptoms.

### 3.3. Between-Group Tests on Sleep Quality, Loneliness, and Alcohol Use

The vape-user group was found to have significantly lower sleep quality as measured by the PSQI. Applying a cut-off of PSQI > 5 (which has been demonstrated to reliably identify the presence of clinically significant insomnia symptoms), revealed that 63.9% of the control group met the criteria, while 77.6% of the vape-user group met the criteria. 

The vape-user group was found to have higher levels of self-reported loneliness (at trend). Further, the vape-user group reported significantly higher levels of alcohol consumption in terms of units consumed per week.

## 4. Discussion

In this study, we investigated the risk factors (in terms of personality characteristics and chronotype) associated with vape use amongst young adults and their links to mental health and sleep quality. Rather than the big five personality traits (which previous work has indicated might not be reliably implicated in vape use risk), we focused on trait rumination, mindfulness, and self-compassion, which have not been studied in relation to vape use. Likewise, there is a paucity of work on links to chronotype and sleep quality, which this study addresses.

Various interesting and novel associations were found. In terms of trait risk factors, the vape use group scored lower on mindfulness, higher on rumination, and lower on self-compassion. These relationships were predicted, as there is some precedent for them in the literature around combustible cigarette use. While a couple of previous studies have examined links between vape use and the Big Five personality traits (since the combustible cigarette literature also implicates these), associations have not been consistent [13]. Thus, the current study adds important novel insights into the traits associated with vape use, pointing to a significant role for mindfulness, rumination, and self-compassion as trait-level factors that might predispose young adults to use vape products.

These findings are in line with the notion that vape use, like cigarette use, might serve as a means to self-regulate negative affect [14]. This negative reinforcement model suggests that a need to avoid negative sensations via self-medication is the most salient motivator of chronic substance use [40].

Mindfulness promotes better affect regulation and the ability to view aversive feelings as transient [16]. Individuals with higher levels of trait mindfulness are therefore less likely to engage in substance use coping behaviours [18,41]. While previous work has shown poorer self-control and emotion regulation abilities amongst young adults who vape [15], this is the first study to show that lower levels of trait mindfulness are present amongst vape users. We also show for the first time that higher levels of trait rumination are present in vape users. Rumination, which consists of repetitive and intrusive thoughts about past negative emotional experiences, is associated with higher levels of distress. Thus, vape use could be a means to self-medicate against this distress amongst high ruminators. Further, there is an inverse relationship between rumination and mindfulness; it is theorised that higher-trait mindfulness allows people to notice ruminative thoughts at an earlier stage and thus disengage from self-perpetuating ruminative thought patterns [42]. Hence, the current findings are in line with a negative reinforcement model of vape use and the theoretical relationships between the rumination and mindfulness constructs.

Self-compassion was also seen to be lower among vape users. Self-compassion enhances emotional resilience by promoting a supportive self-response in times of difficulty. It is conceptually related to mindfulness, since mindfulness enhances awareness of one’s own suffering while self-compassion helps to address it. Studies have shown a close relationship between the mindfulness and self-compassion constructs, which combine to promote emotional well-being [43]. Self-compassion has been linked to better health behaviours, and an intervention study found that training smokers to self-regulate from a self-compassionate stance helped them reduce cigarette smoking [44]. However, there is little work around trait self-compassion and substance use behaviours, and this is the first study to investigate and show a link to vape use.

In line with some previous studies showing poorer mental health amongst vape users, the vape-user group was found to have significantly higher levels of anxiety; nearly all vape users reported clinically significant anxiety symptoms. This is again in line with the negative reinforcement model, suggesting that vaping might be used as a means of self-medication against unpleasant feelings of anxiety in particular. Not all previous studies have shown associations with anxiety, and we found no link to depressive symptoms. This contrasts with some previous studies, although, as noted by a recent review, the literature is inconsistent [11]. However, many studies to date have not used comprehensive mental health measurement scales. For example, King and Reboussin [9] only asked participants if they had received a mental health diagnosis in the previous 6 months, while Conway and Green [8] used single-item measures to assess anxiety and depression. Thus, the current study adds much-needed detail to the literature by employing a well-used measure of anxiety and depression symptoms (the HADS), which has good clinical validity. Findings suggest that elevated anxiety symptoms are associated with vape use; this finding has strong theoretical support from the negative reinforcement model of substance use.

We also investigated how chronotype and sleep quality differ amongst vape users. Evening types tend to report higher substance use; they are more likely to smoke cigarettes [24], but there is a lack of studies on vaping. We found a significant over-representation of evening types amongst vape users: nearly ¾ were evening types as compared to only 40% of non-users. As noted in the introduction, a later sleep time might provide an additional opportunity to vape. Vape users also reported (at trend) higher levels of loneliness; this could be related to eveningness since it has been shown that young adult evening types report lower levels of social support [45]. We also found significant links to sleep quality. While cigarette use has been associated with higher insomnia symptoms [25], there is little work in relation to vaping. The current findings align with those of another study in college students [27], showing that vape users report significantly more sleep difficulties. While the stimulative properties of nicotine might be responsible for impaired sleep, an inverse relationship is also possible, since one study in adolescents showed that sleep deprivation increases vape use risk, perhaps to help compensate for daytime dysfunction [28]. Finally, we also found that vape users report higher levels of alcohol consumption; this is in line with previous work showing that young adult vape users are more likely to have substance use problems [8]. Close correlations between cigarette smoking and alcohol use are well demonstrated in young adult populations: evidence points to overlapping personality risk factors for tobacco and alcohol consumption; studies also suggest that people find tobacco more satisfying when drinking alcohol; and evidence from animal research shows that tobacco use promotes excessive alcohol consumption [46].

### Limitations

Some limitations of the study should be noted. Females were overrepresented in the sample, and the number of males was too low in the vape-user group to allow analysis of possible gender effects. Also, the number of dual-users (cigarettes and vapes) was too low to support an analysis of this subgroup, and thus we opted to remove these individuals from the dataset; the current work therefore does not provide information around the characteristics of dual-users. All measures were based on self-report, and social desirability bias could affect their accuracy. Although this raises ethical issues, objective markers of nicotine use status could potentially enhance accuracy, as could the use of, for example, actigraphy measures for sleep and chronotype. Importantly, the cross-sectional nature of the study does not allow us to draw any conclusions regarding directionality. Thus, while we have shown significant associations with certain trait characteristics and chronotypes, we cannot make claims as to whether these are truly predisposing risk factors. Likewise, with regard to the associations with mental health and sleep quality, we cannot say whether these are a cause or consequence of vape use. Only longitudinal studies can provide insight into the direction of effects. Further, there are interaction effects among these factors that the current study cannot unpick. For example, eveningness is connected with higher stimulant consumption, poorer mental health, and lower sleep quality [23], while poor sleep is both a risk factor and a consequence of depression and anxiety [47]. Thus, there exist complex inter-relationships amongst the variables under study here, and more advanced data collection and analysis methods are required in future studies to delineate these. Future work could also explore the role of other factors not considered in the present study, such as socioeconomic status, access to vape products, and whether perceived stress influences the relationships identified here. Nevertheless, the current study provides important insights and serves as a basis for future investigations.

## 5. Conclusions

Given the high prevalence of vape use amongst young adults in particular, it is important to identify the factors that increase the risk of use and determine how vape use is associated with mental health and other important metrics. This is the first study to investigate and show that young adult vape users have lower trait mindfulness, lower self-compassion, and higher levels of rumination. Vape users were also found to have significantly higher levels of anxiety symptoms and an evening chronotype, with users also reporting poorer sleep quality and a higher level of alcohol use. More loneliness was also reported amongst users (at trend). These novel findings provide important insights into the specific cognitive styles and personality characteristics which serve as risk factors for vape use amongst young adults, as well as the impact of vape use on mental health and sleep quality. The findings have implications for the design and targeting of interventions to reduce vape use prevalence and its consequences, for example through strategies that enhance self-compassion and mindfulness, and combat ruminative tendencies.

## Figures and Tables

**Table 1 healthcare-12-00423-t001:** Means and standard deviations (SD), by group; results of between-group ANOVA.

Measure	Control		Vape-User		
	M	SD	M	SD	*p*-Value
Age	20	1.4	19.9	1.8	0.639
FFMQ	73.41	10.31	69.49	11.11	0.016 *
Rumination	45.47	9.03	49	8.41	0.012 *
rMEQ—Diurnal preference	12.44	3.73	9.23	3.66	<0.001 **
Self-compassion	33.83	7.23	31.02	8.88	0.017 *
HADS–Anxiety	10.25	3.74	12.41	2.93	<0.001 **
HADS–Depression	6.91	3.37	7.41	3.03	0.338
Sleep Quality	7.19	3.26	8.38	3.58	0.023 *
Loneliness	3.48	1.93	4.04	1.68	0.058
Alcohol use (units)	3.54	4.62	7.49	6.26	<0.001 **

Note: For between-group tests based on ANOVA, * indicates significance at the 0.05 level; ** indicates significance at the 0.01 level.

## Data Availability

The data supporting the findings of this study are available upon request.

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
