# Peer review of "Personality Risk Factors for Vape Use amongst Young Adults and Its Consequences for Sleep and Mental Health"

_healthcare, 2024, doi:10.3390/healthcare12040423_

Round 1
Reviewer 1 Report
Comments and Suggestions for Authors
This study by authors Evans and Alkan explores the link between the personality traits and vape use, and relationships with chronotype, sleep quality and mental 11 health, among young adults. While the objectives of the study were of immense interest, the findings have several serious flaw and limitations.
1. A huge difference in the number of subjects in non-smokers versus vaping group [263 vs. 49] may skew data leading to over-presentation and potentially misinterpretation. While the sex-differences in the number of subjects enrolled in two groups were discussed, but may have influenced a lot of parameters studies using the survey.
2. Some of the conclusions drawn on mindfulness, higher renumination rates, and lower self-compassion in the vaping subjects are not novel and have been described earlier for smokers [cigarette and vaping]. Authors identified the factors in this study that cannot be claimed to be truely predisposing risk factors.
3. The most striking difference was in the Alcohol users between two groups. However, authors neglected to explain or discuss this further in light of the current dogma or understanding.
4. Most of the conclusion drawn are either already established for cigarette smokers/vaping and therefore lacks novelty.
Author Response
We thank the reviewer for taking the time to read the manuscript, and we value their insightful comments and suggestions.
1. A huge difference in the number of subjects in non-smokers versus vaping group [263 vs. 49] may skew data leading to over-presentation and potentially misinterpretation. While the sex-differences in the number of subjects enrolled in two groups were discussed, but may have influenced a lot of parameters studies using the survey.
Indeed, our sample comprised ~16% vape users. This percentage is in line with previous data from college-age populations (please see eg. https://www.sciencedirect.com/science/article/pii/S0191886919305379#bb0200 and https://www.sciencedirect.com/science/article/pii/S0376871615000046). We apologise for the oversight of not including information about the assumption checks – this has now been added to Methods. The data met assumptions of normality and homogeneity of variances and was therefore suitable for parametric ANOVA (equal group sizes is not a requirement for ANOVA). Please note that a similar approach has taken by previous studies, see eg. https://www.ncbi.nlm.nih.gov/pmc/articles/PMC7317082/ , also with mismatched group Ns. Given that the data met the assumptions for ANOVA, and the size of the vape user group was not small (and indeed comparable to previous similar studies), we would argue that the results are robust (we would also like to point out that the effects seen were present at p values well below the 0.05 threshold, and indeed some were significant at p<0.001). Regarding gender, this is a common issue in the psychology literature (eg. both of the aforementioned studies also had an overrepresentation of female students) but we kindly point out that in our study, there was no significant difference in the gender makeup of our vape vs non-vape groups. Thus, as we point out in the discussion, the overrepresentation of females should be regarded as a limitation (in terms of generalisability)- but it is unlikely to underlie the between-group findings we present as the two groups were comparable in terms of gender balance.
2. Some of the conclusions drawn on mindfulness, higher renumination rates, and lower self-compassion in the vaping subjects are not novel and have been described earlier for smokers [cigarette and vaping]. Authors identified the factors in this study that cannot be claimed to be truely predisposing risk factors.
Thank you for this. As we point out, while most of the factors have indeed been studies in relation to cigarette smoking, this is the first time some of them have been studied in relation to vaping - and the purpose of the paper is to see which/whether are also applicable to vaping. We performed a thorough literature review of the vaping lit and we discuss which have been tested before (or otherwise). Further, we already clearly discuss the limitations of the study in terms of inability to draw causal inferences.
3. The most striking difference was in the Alcohol users between two groups. However, authors neglected to explain or discuss this further in light of the current dogma or understanding.
Thank you for this - we have expanded our discussion of this point. However, we would like to point out that while this is an interesting finding, it is one that has been consistently shown in the literature. Thus, we focus the discussion mostly on the more novel findings.
4. Most of the conclusion drawn are either already established for cigarette smokers/vaping and therefore lacks novelty
Please see our response to point 2.
Reviewer 2 Report
Comments and Suggestions for Authors
This study explores the personality traits linked to vape use and relationships with chronotype, sleep quality, and mental health among young adults. The study found that the vape user group scored lower on mindfulness, higher on rumination, and lower on self-compassion. Vape users were more likely to be evening types and to have significantly lower sleep 15 quality and higher anxiety symptoms.
The paper is well-written and organized. However, there are a few issues that have to be addressed:
- At the beginning of the second paragraph in the introduction, the authors said, 'Various studies have examined links to mental health.' There is a need to clarify it because it is ambiguous.
- In the methods section, the authors stated that they used ANOVA. It would be better for them to specify the type (one-way or 2-way).
- In the methods section, the authors have to specify the dependent and independent variables used in the study.
- What sampling method was used in the data collection?
Author Response
We thank the reviewer for taking the time to read the manuscript, and we value their insightful suggestions and positive comments.
- At the beginning of the second paragraph in the introduction, the authors said, 'Various studies have examined links to mental health.' There is a need to clarify it because it is ambiguous. Thank you for this- we have revised.
- In the methods section, the authors stated that they used ANOVA. It would be better for them to specify the type (one-way or 2-way). Thank you for this- we have clarified.
- In the methods section, the authors have to specify the dependent and independent variables used in the study. Thank you - we have added.
- What sampling method was used in the data collection? Thank you - we have now specified Opportunity sampling in Methods.
Reviewer 3 Report
Comments and Suggestions for Authors
The authors explored an interesting topic related to the use of e-cigarettes among young adults. However, both the title and abstract suggest that personality traits will be the main independent variables, but these are barely addressed in the research beyond a mention in the theoretical part. Consequently, the title of the article needs to be changed as it does not match the content.
The theoretical introduction is coherent and concise; however, it lacks a justification for why the authors chose specific questionnaires to measure the independent variables, such as mindfulness, depression, rumination, morningness-eveningness, etc., among the numerous options available. As each measurement tool is based on a specific theory, the lack of a universal solution for psychological questionnaires requires clarification.
The research aimed to better understand e-cigarette users, but the study raises doubts about the validity of the results and the conclusions drawn. The main concern lies in the proportions of the sample studied - 263 people made up the reference group, while only 49 belonged to the group of e-cigarette smokers. In other words, the group of smokers represents less than 20% of the control group, which indicates a significant imbalance.
There are also concerns about the data analysis methods used. Why use ANOVA? In the case of two independent groups, simpler solutions are available such as the t-Student test or its non-parametric equivalent. Although the use of ANOVA is not incorrect under certain assumptions, there is a lack of information about the compatibility of the distributions of the variables with normality in the compared subpopulations or the homogeneity of the variances. The equivalence of the groups compared is clearly not met.
Analytically, one possible solution would be to use the bootstrap method. The other solution is to randomly select a subset of participants (about 50-60 individuals) from a larger reference group whose results would be compared with the group of e-cigarette users. However, the most sensible solution would be to study more individuals who actually use e-cigarettes to ensure that the conclusions drawn from the analyses are less like fortune-telling.
Author Response
We thank the reviewer for taking the time to read the manuscript, and we value their detailed comments and suggestions, which we respond to below.
The authors explored an interesting topic related to the use of e-cigarettes among young adults. However, both the title and abstract suggest that personality traits will be the main independent variables, but these are barely addressed in the research beyond a mention in the theoretical part. Consequently, the title of the article needs to be changed as it does not match the content.
Thank you for this. Indeed, we were interested in studying the differences between vape/non-vape users, on personality variables (Mindfulness, Rumination, Self-compassion), sleep, and mental health. Mindfulness, Rumination and Self-compassion were compared between-groups and some novel and interesting differences identified, these are discussed at length. Therefore we would suggest that they are a main focus of the paper, and thus the title is appropriate.
The theoretical introduction is coherent and concise; however, it lacks a justification for why the authors chose specific questionnaires to measure the independent variables, such as mindfulness, depression, rumination, morningness-eveningness, etc., among the numerous options available. As each measurement tool is based on a specific theory, the lack of a universal solution for psychological questionnaires requires clarification.
Thank you for this. The reviewer is correct to note that other alternatives exist for measuring these IVs. We chose our instruments based on their reliability and validity: the questionnaires we selected are all very well established with high levels of reliability/validity. The reviewer is also correct to note the possibility of different theoretical underpinnings – this is true for mindfulness and rumination (for mental health and chronotype, the HADS and rMEQ are standard scales and not theory-specific, and each has been used by thousands of previous studies). The FFMQ was selected because it is a unifying and comprehensive scale that integrates the conceptualizations of mindfulness from five previous (validated) mindfulness scales. The RRQ was selected due to its high levels of construct validity, reliability, and because it is one of the most widely-used scales for rumination. We have added detail to explain this in the Methods section. Regarding the reviewer’s point about “a universal solution for psychological questionnaires” – the FFMQ was formulated as a universal approach to mindfulness, but for the other IVs, no “universal” approach exists so we selected scales based on reliability/validity and how well established they are.
The research aimed to better understand e-cigarette users, but the study raises doubts about the validity of the results and the conclusions drawn. The main concern lies in the proportions of the sample studied - 263 people made up the reference group, while only 49 belonged to the group of e-cigarette smokers. In other words, the group of smokers represents less than 20% of the control group, which indicates a significant imbalance. There are also concerns about the data analysis methods used. Why use ANOVA? In the case of two independent groups, simpler solutions are available such as the t-Student test or its non-parametric equivalent. Although the use of ANOVA is not incorrect under certain assumptions, there is a lack of information about the compatibility of the distributions of the variables with normality in the compared subpopulations or the homogeneity of the variances. The equivalence of the groups compared is clearly not met.
Indeed, our sample comprised ~16% vape users. This percentage is in line with previous data from college-age populations (please see eg. https://www.sciencedirect.com/science/article/pii/S0191886919305379#bb0200 and https://www.sciencedirect.com/science/article/pii/S0376871615000046). We apologise for the oversight of not including information about the assumption checks – this has now been added to Methods. The data met assumptions of normality and homogeneity of variances and was therefore suitable for parametric ANOVA (equal group sizes is not a requirement for ANOVA). Given this, we would argue that ANOVA is most appropriate for testing the hypotheses (rather than a serious of multiple t-tests). Please note that a similar approach has taken by previous studies, see eg. https://www.ncbi.nlm.nih.gov/pmc/articles/PMC7317082/ , also with mismatched group Ns.
Analytically, one possible solution would be to use the bootstrap method. The other solution is to randomly select a subset of participants (about 50-60 individuals) from a larger reference group whose results would be compared with the group of e-cigarette users. However, the most sensible solution would be to study more individuals who actually use e-cigarettes to ensure that the conclusions drawn from the analyses are less like fortune-telling.
Thank you for these suggestions. However, as stated above, given that the data met the assumptions for ANOVA, and the size of the vape user group was not small (and indeed comparable to previous similar studies), we would argue that the results are unlikely to be due to chance. We would also like to point out that the effects seen were present at p values well below the 0.05 threshold, and indeed some were significant at p<0.001. This suggests a good level of confidence in the robustness of our findings, thus justifying the conclusions drawn.
Reviewer 4 Report
Comments and Suggestions for Authors
The current manuscript from Dr.s Evans and Elkan focused on three personality risk factors, namely mindfulness, rumination, and self-compassion, as predictors of e-cigarette use amongst a sample of young adults. Additionally, authors also explored the role of chronotype in vape use.
Authors considered two groups, namely control non-users and vape-users. Significant differences were noted between the two groups in terms of the scores of Five Facet Mindfulness Questionnaire (FFMQ), Hospital Anxiety Depression Scale (HADS), Rumination Reflection Questionnaire (RRQ), Reduced Morningness Eveningness Questionnaire (rMEQ), alcohol use, and sleep quality.
Although the observations are interesting and the information is very critical, some issues with the study design and data analyses should be addressed.
Suggestions
1. The most critical issue with the study is the presence of 14 vape users, who are also cigarette smokers. The consensus is that these users should be identified as “dual users”. Authors should consider performing an additional analysis considering three groups: non-users, vape-users, and dual-users.
2. Authors are requested to expand their perception on the possible effect of large difference in participant numbers between the two groups on the statistical analysis. Currently, the control non-user group has 263 participants, whereas the vape-user group has 49 subjects. Will it be more appropriate to compare between age- and sex-matched individuals from both groups?
3. Since no tests were performed to verify the tobacco-use status of the participants, the use pattern is dependent on self-report, this is a limitation of the study. Please expand discussion on this.
4. Although the cigarette smokers were removed from the study, it will be very interesting if the authors can include a brief discussion on the characteristics of these participants, compared to matched control non-users.
5. Authors are requested to expand the limitations section of the manuscript, at their own discretion.
6. Please edit the conclusion section to describe what the observations mean, in a language that readers unrelated to the research field may understand easily.
7. Please expand the description of the statistical analysis strategy adopted in this study. The current description needs more information.
Other comments
1. Introduction, “Electronic cigarettes (e-cigarettes, ‘vapes’) are battery-operated products that use a nicotine-containing solution to deliver nicotine via inhalable aerosol.” Authors should mention nicotine-free e-liquids too.
2. Authors are requested to re-write the last paragraph of the introduction where they describe the study objectives. More precise language will be very helpful.
3. Authors may remove the first word “Nevertheless” from the conclusions section.
4. If available, please include the types of e-cigarettes used by the participants. Any information will be very supportive, including generation of device, freebase or salt nicotine, flavors used by the participants, to name a few.
Comments on the Quality of English LanguageMinor editing should be considered by the authors.
Author Response
The most critical issue with the study is the presence of 14 vape users, who are also cigarette smokers. The consensus is that these users should be identified as “dual users”. Authors should consider performing an additional analysis considering three groups: non-users, vape-users, and dual-users.
We thank the reviewer for this suggestion - indeed we noted that previous studies performed subgroup analysis of dual-users. We opted not to explore this group, as the N was so low – we wanted to ensure that the conclusions we draw are reliable and valid, and performing analysis on an N of 14 would mean that any findings are subject to a high risk of statistical error. Therefore we opted to remove these participants completely (also, the main focus of the paper is vape use specifically, rather than dual use). This is now discussed in the limitations section.
Authors are requested to expand their perception on the possible effect of large difference in participant numbers between the two groups on the statistical analysis. Currently, the control non-user group has 263 participants, whereas the vape-user group has 49 subjects. Will it be more appropriate to compare between age- and sex-matched individuals from both groups?
Indeed, our sample comprised ~16% vape users. This percentage is in line with previous data from college-age populations (please see eg. https://www.sciencedirect.com/science/article/pii/S0191886919305379#bb0200 and https://www.sciencedirect.com/science/article/pii/S0376871615000046). We apologise for the oversight of not including information about the assumption checks – this has now been added to Methods. The data met assumptions of normality and homogeneity of variances and was therefore suitable for parametric ANOVA (equal group sizes is not a requirement for ANOVA). The groups were well matched on age and gender (as all were undergraduates from the same course), an alternative analysis strategy of comparing between age- and sex-matched individuals from both groups carries risk of selection error/bias, and thus we would suggest that the ANOVA approach we employed is more appropriate (please note that a similar approach has taken by previous studies, see eg. https://www.ncbi.nlm.nih.gov/pmc/articles/PMC7317082/ , also with mismatched group Ns).
Since no tests were performed to verify the tobacco-use status of the participants, the use pattern is dependent on self-report, this is a limitation of the study. Please expand discussion on this.
Thank you for this, we have added.
Although the cigarette smokers were removed from the study, it will be very interesting if the authors can include a brief discussion on the characteristics of these participants, compared to matched control non-users.
Please see our response to the previous point.
Authors are requested to expand the limitations section of the manuscript, at their own discretion.
Thank you – we agree that this is necessary, we have expanded.
Please edit the conclusion section to describe what the observations mean, in a language that readers unrelated to the research field may understand easily.
Thank you – we have made significant edits to the conclusion, for clarity.
Please expand the description of the statistical analysis strategy adopted in this study. The current description needs more information.
Apologies for the lack of detail, which we have now addressed.
Other comments
Introduction, “Electronic cigarettes (e-cigarettes, ‘vapes’) are battery-operated products that use a nicotine-containing solution to deliver nicotine via inhalable aerosol.” Authors should mention nicotine-free e-liquids too.
This has been added.
Authors are requested to re-write the last paragraph of the introduction where they describe the study objectives. More precise language will be very helpful.
Thank you – we have made significant edits to improve clarity.
Authors may remove the first word “Nevertheless” from the conclusions section.
Amended.
If available, please include the types of e-cigarettes used by the participants. Any information will be very supportive, including generation of device, freebase or salt nicotine, flavors used by the participants, to name a few.
Unfortunately, as described in Methods, we did not collect information regarding the type of e-cigarettes used by the participants.
Reviewer 5 Report
Comments and Suggestions for Authors
The paper focuses on the association between personality traits and vaping in young people. The authors explore also the association between vaping and chronotype, sleep quality and mental health of young adults. The relatively small sample size of the risk groups allow only descriptive analysis to be applied. Nevertheless, the results show important dependencies between the studied factors and vaping. The identified association with mental health problems and worsened quality of sleep among vapers can serve as a reference point for interventions, aiming to promote healthy lifestyles and to reduce the use of nicotine products among young people.
My suggestion to the authors is to include data/statistics of the prevalence of smoking and vaping among young people and to comment on and to add references about the observed trends in related to an increasing use of e-cigarettes among them. Another suggestion is to add a paragraph on the future directions of study. Apart from personality traits, sociodemographic, environment and lifestyle factors are also associated with the risk of vaping, i.e. exposure and availability of nicotine products, SES and family background, perceived stress, etc. Some of these factors could be explored in as correlates of vaping that would upgrade the present analysis.
Author Response
We thank the reviewer for their positive comments about our manuscript, and very much appreciate their suggestions. As recommended, we have added some additional details in the Introduction about prevalence rates. We have also added to the Discussion section some possibilities regarding future work (please see Limitations section).
Round 2
Reviewer 4 Report
Comments and Suggestions for Authors
Authors addressed the comments.
Comments on the Quality of English LanguageMinor editing.